# Fast and powerful genome wide association of dense genetic data with high dimensional imaging phenotypes

Habib Ganjgahi[1,2], Anderson M. Winkler [3,4], David C. Glahn[5], John Blangero[6], Brian Donohue[7], Peter Kochunov[7] & Thomas E. Nichols [3,8,9]

Genome wide association (GWA) analysis of brain imaging phenotypes can advance our understanding of the genetic basis of normal and disorder-related variation in the brain. GWA approaches typically use linear mixed effect models to account for non-independence amongst subjects due to factors, such as family relatedness and population structure. The use of these models with high-dimensional imaging phenotypes presents enormous challenges in terms of computational intensity and the need to account multiple testing in both the imaging and genetic domain. Here we present a method that makes mixed models practical with high-dimensional traits by a combination of a transformation applied to the data and model, and the use of a non-iterative variance component estimator. With such speed enhancements permutation tests are feasible, which allows inference on powerful spatial tests like the cluster size statistic.

[1] Department of Statistics, University of Oxford, Oxford, UK. [2] Medical Research Council Harwell Institute, Harwell, UK. [3] Wellcome Centre for Integrative Neuroimaging, University of Oxford, Oxford, UK. [4] Big Data Analytics Group, Hospital Israelita Albert Einstein, São Paulo, SP, Brazil. [5] Department of Psychiatry, Yale University School of Medicine, New Haven, CT, USA. [6] South Texas Diabetes and Obesity Institute, University of Texas Rio Grande Valley School of Medicine, Brownsville, TX, USA. [7] Maryland Psychiatric Research Center, Department of Psychiatry, University of Maryland School of Medicine, Baltimore, MD, USA. [8] Oxford Big Data Institute, Li Ka Shing Centre for Health Information and Discovery, Nuffield Department of Population Health, University of Oxford, Oxford, UK. [9] Department of Statistics, University of Warwick, Coventry, UK. Correspondence and requests for materials should be addressed to T.E.N. (email: thomas.nichols@bdi.ox.ac.uk)

Genome-wide association studies (GWAS) of neuroimaging data can advance our understanding of human brain by discovering genetic variants associated with normal and disorder-related phenotypic variance in brain structure and function[1–6]. Quantitative brain phenotypes are based on structural images (i.e. brain volume, cortical thickness, white matter integrity) or functional images (brain response to particular cognitive task or resting state). Genetic association at a hundred thousand locations in the human brain present immense statistical challenges including: low statistical power, need for multiple comparisons correction and like, other association studies, correction for population structure, a term that encompasses cryptic/family relatedness and population stratification.

In the GWA studies of unrelated individuals, non-independence due to latent population stratification or unknown (cryptic) relatedness[7,8] is generally thought to be a confounding factor that can lead to excessive false positives when ignored. This type of non-independence has been studied throughly in the recent GWA era[9–13]. While genomic data can be used to control for population stratification by including the top principal components as fixed effect covariates in a linear regression model[14], usually individuals with close estimated relatedness from identity-by-state (IBS) matrix or different ethnicities are excluded from the study sample. This might not be a problem in genetic studies with six digit sample sizes, but may make substantial differences in GWA studies with neuroimaging phenotypes where sample size is much smaller. Also, even in a carefully designed GWA study, it is hard to avoid spurious associations because of population structure; in particular, it is likely that in studies with large sample sizes, such as the UK biobank possess some level of population structure. Although the emergence of large scale neuroimaging consortia like ENIGMA or CHARGE can help to conduct well-powered genetic association studies through meta analysis framework, still it is crucial to use a powerful statistical method at the site level. Hence, there is a compelling need for analytical techniques that addresses these challenges.

Linear mixed effect models (LMMs) using molecularly derived empirical relatedness measures have grown in use recently for both studies of related and unrelated individuals, since they do not require self-reported biological relatedness, providing a framework where such complexities are automatically accounted for. LMMs have been used as an alternative to ordinary least squares (OLS), providing a mechanism to model trait variance explained by a genetic relationship matrix (GRM); such variance captures the genome-wide similarity between "unrelated" individuals by modeling it as a random effect[15–26]. It has been shown that the correction for the problem of latent population structure in GWA with an LMM is both effective and power preserving[25–27].

These merits motivate the application of LMM for genetic association with high-dimensional imaging phenotypes. However, fitting LMM at each voxel in the brain is computationally intensive or even intractable at the voxel level since variance component estimation relies on likelihood function optimization using numerical methods. Moreover, search for genetic association across the genome at different locations with imaging phenotypes requires intense multiple testing corrections both for number of elements in an image and number of markers. Whether the association analysis is conducted for a few regions of interest (ROIs) or every voxel, naive application of Bonferroni correction for number of hypothesis testing in the image with usual GWA P-value leads to conservative statistical inference procedure as it ignores complex spatial dependence in the imaging phenotypes. Despite the many analytical techniques that have been developed to accelerate the GWA with LMM, these advances do not eliminate problems related to numerical optimization nor multiple testing problem. Furthermore, commonly used spatial inference tools, like cluster size[28] or threshold-free cluster enhancement (TFCE) statistics[29], depend on resampling-based inference methods to ensure valid control of false positives[30,31]. Familywise error rate (FWE) correction, controlling the chance of one or more false positives across the whole set (family) of tests[32] requires the distribution the maximum statistic, can be computed for either voxels, ROIs, clusters or TFCE with a permutation test[33], a standard tool to conduct inference in neuroimaging.

This paper makes two major contributions to reduce the complexity of LMM for genetic association with imaging phenotypes. First, the computational cost of variance component estimation is reduced by using a non-iterative one-step random effect estimator[34]. Second, the complexity of association testing is dramatically decreased by projecting the model and phenotype to a lower dimension space, combined with use of a score statistic for association testing. This projection is based on the eigenvectors of the GRM adjusted for the fixed effect nuisance terms[16,35]. In this setting, the projected phenotype likelihood function is equivalent to that used with restricted maximum likelihood (REML) of the LMM (Method), going forward we call this approach *simplified REML*[35]. While both models have the same statistical properties, our particular projection provides several computational benefits that dramatically reduces LMM complexity: (i) The diagonalized covariance allows a non-iterative one-step variance component estimator[34], taking the form of a weighted regression of squared projected data on eigenvalues of the GRM adjusted for nuisance fixed effect terms, an approach that we call weighted least squares REML (*WLS-REML*); (ii) The regression form of our estimator is easily vectorized, meaning that many image elements and SNPs can be tested in a single and fast computational test in several high-level programming languages (Method); (iii) Finally, the simplicity and fast computation of the score test statistic makes permutation testing feasible, allowing exact, non-parametric control over the FWE, accounting for the number of tests conducted over all image elements and genetic markers; we define two permutation schemes, free and constrained, where in the latter case the permutation is confined to exchangeability blocks defined based on the eigenvalues distribution.

## Results

**Computational complexity**. The reduced computational complexity of our method represents a significant advance over existing methods. The complexity of LMM association has two components, one for the variance component estimation, the other is for fixed effect parameter estimation and test statistic computation. For a GWA over $S$ markers and $V$ imaging phenotype elements on $N$ individuals, the variance component likelihood optimization complexity of FaST-LMM[19,35] which, to the best of our knowledge, is the fastest implementation of LMM is $O(N^3 + INV)$, where $I$ is the average number of iterations, while for WLS-REML the random effect estimator (see Methods) it is $O(N^3 + NV)$ (the common $O(N^3)$ term is the time complexity of the GRM eigendecomposition). More critically, the estimation and test statistic computation complexity of FaST-LMM is $O(SPN^2V)$, where $P$ is the number of nuisance fixed effects, while for WLS-REML (Eq. (16)) this is $O(SNV)$, a substantial reduction for imaging phenotypes when number of image elements $V$ is much bigger than the sample size $N$. Even for a single trait GWA ($V = 1$), our proposed projection reduces the association (Eq. (13)) complexity to $O(SN)$ which is significantly less than FaST-LMM for large sample GWA.

In our previous work[34], we introduced the WLS-ML random effect estimator that exploits a one-step optimization approach

combined with eigen-rotation of phenotype and model (see Method for more details). The non-iterative estimator has a simple form, with variance components and fixed parameters each estimated by weighted least-squares regressions. In this paper, we evaluate our non-iterative ML and REML estimators (WLS-ML and WLS-REML) with their fully converged counterparts (Full ML or Full REML), comparing score, likelihood ratio (LRT) and Wald tests on intensive simulation studies; the two permutation schemes are also compared. Direct comparisons are made between FaST-LMM and our score test with WLS-REML using simulation and real data.

**Simulation results**. Intensive simulation studies are conducted to evaluate the proposed methods for association estimation and testing. The aim of the first study is to compare fully converged and one-step random effect estimators based on the simplified ML and REML functions. In the second study, the performance of various test statistics for the association testing are compared using a fully converged or one-step random effect estimators for ML and REML functions (see Supplementary Note 1 for details). Finally, we compare FaST-LMM to our preferred test, the score test based on the simplified REML function, using both false positive error rates and empirical power using simulated genetic markers (see Supplementary Note 2 for details).

Simulation results on the accuracy of genetic random effect $(\sigma_A^2)$ estimation shows that the non-iterative one-step approaches are similar to their fully converged counterparts (Supplementary Figure 1), using either likelihood or restricted likelihood functions. When the data are independent $(\sigma_A^2 = 0)$, the methods are indistinguishable in terms of bias and mean squared error (MSE). When $\sigma_A^2 > 0$, the fully converged methods have less bias, but the difference is modest in absolute value; in terms of MSE, the non-iterative one-step methods have just slightly worse performance. The first simulation also shows good performance of fixed effect $(\beta_1)$ estimation (Supplementary Figure 2). Both the non-iterative one-step and fully converged have similar bias and MSE, with WLS-REML again closely following fully converged REML.

Simulations show that the false positive rates for the fixed effect score test for $H_0:\beta_1 = 0$ (Supplementary Figure 3a) are nominal; for both simplified ML or REML functions, for all simulation settings considered, test statistic type and type of random effect estimator, the false positive rates lay within the Monte Carlo confidence interval (MCCI) (see Supplementary Figures 4a, b).

The simulation results on the power of score test reveal negligible differences between the random effect estimation methods (Supplementary Figure 3b). Similar findings are obtained for the power of LRT and Wald tests (Supplementary Figures 4c, d). Like the parametric approach, we found that both permutation schemes, free or permutation within exchangeability blocks, control the false positives at the nominal level 5% (Fig. 1a and Supplementary Figures 5a, b), and could provide nearly equivalent power Fig. 1b, Supplementary Figures 5c, d) for all statistics either based on the simplified ML or REML functions. However, for all test statistics and $\sigma_A^2$, the free permutation scheme is slightly more powerful than the constrained permutation test when a kinship matrix is used.

Simulations show that the null distribution of the score test for $H_0:\beta_1 = 0$ based on the simplified models using the fully converged and non-iterative variance component estimators are valid and indistinguishable (Fig. 2 and Supplementary Figure 6). However, we stress that the latter is much faster to calculate. Based on all of these results, we selected the score test based on the simplified REML function as the computationally most efficient test to be considered for genome-wide simulations and real data analysis.

Genome wide simulations were conducted to compare the parametric P-values from FaST-LMM and the score test based on the simplified REML using non-iterative variance component estimator in terms of false positives and power. The simulation results reveal that both approaches provide overall valid error rates (FaST-LMM = 4.94% and the WLS-REML score test = 4.89%, Fig. 3a). Power simulation shows that FaST-LMM and the score test have largely similar power (FaST-LMM = 15.25%, WLS-REML score test = 15.22%), however, FaST-LMM is slower (Fig. 3b). Despite reasonable concordance of P-value and fixed effect parameter estimates $(\beta_1)$ between FaST-LMM and simplified REML (Supplementary Figure 7), FaST-LMM's estimates of parameter estimate variance $(\text{var}(\hat{\beta}))$ exhibits some systematic bias (Supplementary Figure 8).

The final simulation study evaluates controlling for a heritable fixed effect nuisance covariate in the null simulation setting when there is neither a SNP effect nor a covariate effect on the phenotype. Although the LMM can accommodate fixed effect nuisance terms, we compare to an alternate approach where nuisance covariates are regressed out in advance and LMM is fitted to the residualized phenotypes for GWA. We note that imaging association studies routinely use intracranial volume (ICV) as a nuisance covariate[1,2], and ICV is well known to have heritability as large as 0.8[36,37]. Figure 4 compares performance of FaST-LMM, EMMAX and the score test based on the simplified REML function using non-iterative random effect estimator (NINGA). The simulation results show that the parametric P-value from all approaches when the nuisance covariate is included in the LMM is valid (Fig. 4 left panel). However, the null distribution of P-values from LMM fitted on residualised phenotypes can be conservative (Fig. 4 right panel).

**Association analysis of fractional anisotropy (FA) data**. We performed GWA of whole brain FA data, using a whole brain parcellation of 42 ROIs, as well as a voxel-wise analysis for 53,458 voxels (332 subjects, 1,376,877 SNPs; for full details see Methods), comparing the WLS-REML score test with the fully converged random effect estimators with FaST-LMM. We also evaluate the use of OLS with MDS as nuisance fixed effects regressors for control of population structure in GWA with unrelated individuals.

The random effect estimators, one-step and fully converged REML are compared directly in Fig. 5 with a scatter plot, showing an apparent trade-off between accuracy and running time as the non-iterative method has lower estimates of $\sigma_A^2$ for some regions.

Even with the tendency for genetic variance to be under-estimated with the non-iterative method, the association statistic show remarkable concordance, with both approaches having almost the same performance (Fig. 6). FaST-LMM comparisons with the score test using the simplified REML function shows slightly larger statistics consistently for all ROIs, regardless of random effect estimation method (Supplementary Figures 10 and 11). Furthermore, comparing different approaches genomic control shows that regardless of random effect estimation method, the score test based on the simplified REML has smaller genomic control values than OLS with MDS nuisance regressors for all ROIs consistently. The genomic control of OLS with MDS nuisance regressors is poor, while the score test using both fully converged, one-step estimators and FaST-LMM have similar values close to unity (Supplementary Figure 9).

Figure 7 compares QQ-plot of association statistics between our model, FaST-LMM and OLS with MDS. These plots show either an identical distribution or slightly larger values for the OLS approach; however, the OLS approach has poor genomic control (Supplementary Figure 9) and after adjustment we get essentially identical results (Supplementary Figures 12 and 13).

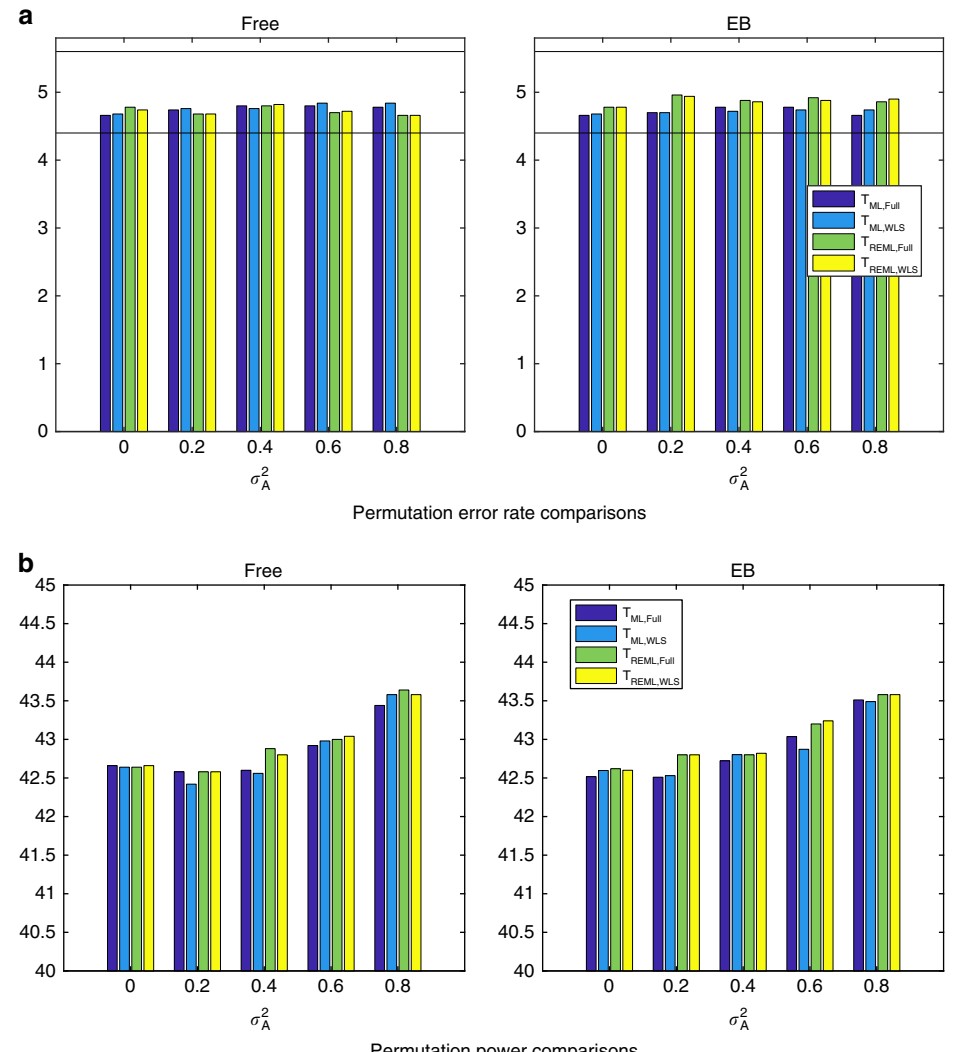

**Fig. 1** Comparison of rejection rates for the proposed fixed effects permutation inference, two different permutation schemes (Simulation 2). Results shown for the null $\beta_1 = 0$ (**a**) and alternative (**b**) for a 5% nominal level based on simulation using a GRM from 300 unrelated individuals and 5000 realizations and 500 permutations each realizations; left column shows results for the free permutation scheme, right for the exchangeability-block constrained method. Monte Carlo confidence interval is (4.40, 5.60%). For non-iterative and fully converged, both permutation schemes could control the error rate at the nominal level, however free permutation is slightly more powerful than the constrained permutation

A permutation test was used to find FWE-corrected *P*-values for 42 ROIs and 1,376,877 SNPs to assess association significance. Among the $42 \times 1,376,877 \approx 57$ million statistics, eight passed the permutation based FWE threshold $(\chi_1^2 = 34.72)$. Application of a Bonferroni correction for 42 tests to the usual GWA alpha level $(5 \times 10^{-8})$ yields to a more stringent threshold $(\chi_1^2 = 36.98)$ where only one association survives, indicating the potential improved power from a permutation-based inference that accounts for dependency among the tests (Fig. 8). An alternate, approximate approach involves computing the effective number of independent tests among the 42 ROIs; while there is no unique definition for the number of independent tests, we used one approach based on eigenvalues of phenotype correlation matrix[38]; this gave an effective test count of 18.30. Application of a Bonferroni correction for 18 independent tests to the usual GWA alpha level $(5 \times 10^{-8})$. This Bonferroni threshold with effective number of tests yields a slightly more stringent threshold $(\chi_1^2 = 35.37)$, but finds the same eight association statistics as found with permutation.

Finally we performed voxel-wise genome-wide association analysis of 53,458 voxels with 1,376,877 SNPs, using our

proposed WLS-REML score test for association. Cluster-wise inference was performed on each spatial association map; we used a threshold corresponding to a $\chi_1^2$ *P*-value of 0.01 to create clusters, and 1000 permutations were used to compute the maximum distribution of cluster size over space and SNPs, offering FWE control over the entire search space; voxel-wise FWE thresholds were also computed. The level 5% FWE-corrected voxel-wise statistic threshold was 66.42, producing six significant association out of 84 billion tests. The 5% FWE corrected cluster size threshold is 7370 but no SNP's statistic map had a cluster exceeding this value; the largest observed cluster size is 6648, which had a image-wide, genome-wide FWE-corrected cluster size *P*-value of 0.09. We note that the effective number of independent tests is not applicable to cluster-wise inference[30] and does not scale to voxel-wise inference. Also, it depends on an arbitrary GWAS threshold that depends on the chip used.

**Benchmarking and running times**. We compared running time of our WLS-REML score test to that of FaST-LMM, which to our knowledge is the fastest implementation of LMM. The

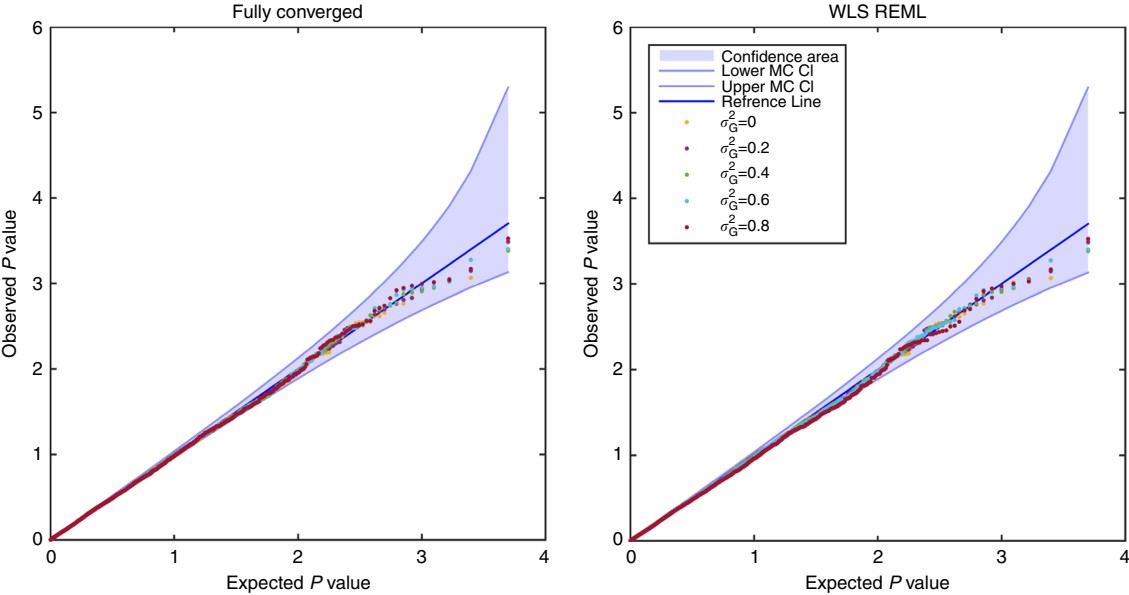

**Fig. 2** Comparison of random effect estimators for fixed effect inference (Simulation 3). Results shown for the distribution of null ($\beta_1 = 0$) parametric P-values from the fixed effects score statistic derived from the simplified REML function using fully converged (left) and non-iterative (right) random effect estimator, for the GRM from 300 unrelated individuals. There is no apparent difference between the two random effect estimators, and both are consistent with a valid (uniform) P-value distribution. Confidence bounds created with the results of ref.[59] where ordered P-values follow beta distribution

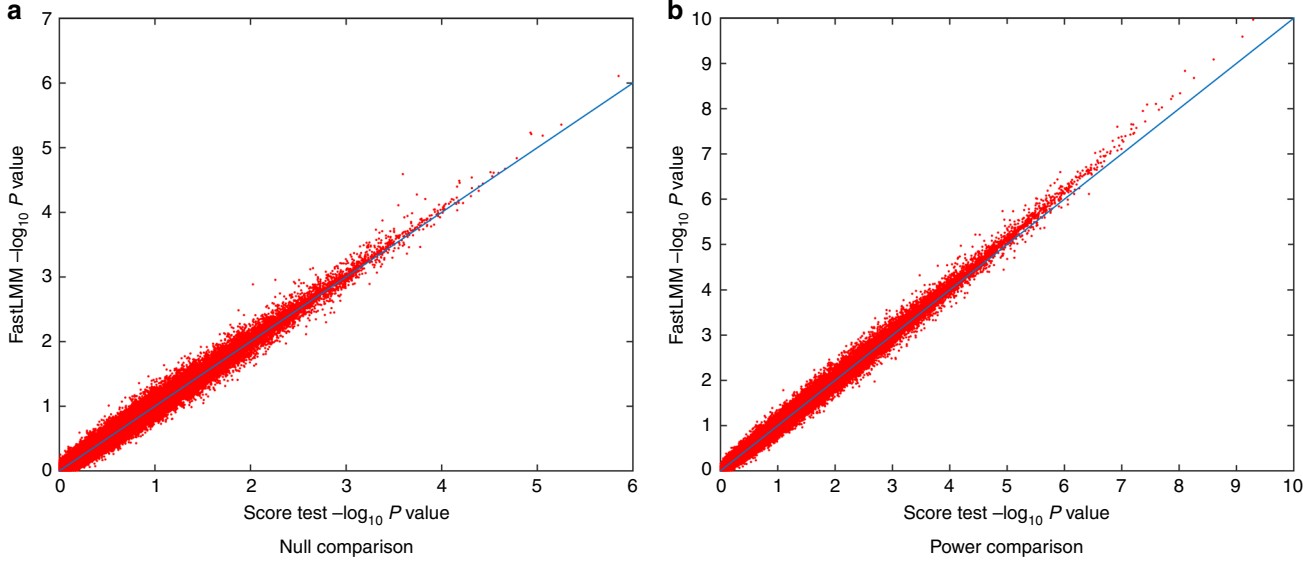

**Fig. 3** Comparison of parametric P-values from FaSTLMM and the proposed method for genome-wide association analysis based on simulated data (Simulation 4). Results shown for null data (**a**) and data with signal (**b**) using FaST-LMM's LRT and our score test based on one-step optimization of the simplified REML function, using 100 random markers and 5000 realizations. The overall error rates for FaST-LMM and the score test are 4.94% and 4.89%, respectively, for nominal 5%, where the Monte Carlo confidence interval is (4.40%,5.60%). While overall power is largely similar for both approaches (for FaST-LMM 15.25% and, and our method 15.22%), FaST-LMM is 200-fold slower

comparison was done using simulated and read data with a Intel (R) core(TM) 3.4 GHz i7-2600 CPU and 16 GB RAM. Parametric association testing of 5000 phenotypes with 6000 simulated markers using a sample of 300 individuals took 1 h with FaST-LMM, however, our implementation of the score test (Eq. (14-15)) only took 3 s. On real data, parametric whole genome association on 42 ROIs, required 756 min using FaST-LMM while our approach took only 2 min.

## Discussion

Neuroimaging genetics has moved from establishing a heritable phenotypes to finding genetic markers that are associated with imaging phenotypes. Despite emerging world-wide consortia to boost GWA studies power using the largest possible sample sizes, there is a compelling need for powerful and computationally efficient analytic techniques that control for population structure at the site level. Moreover, GWA studies with neuroimaging

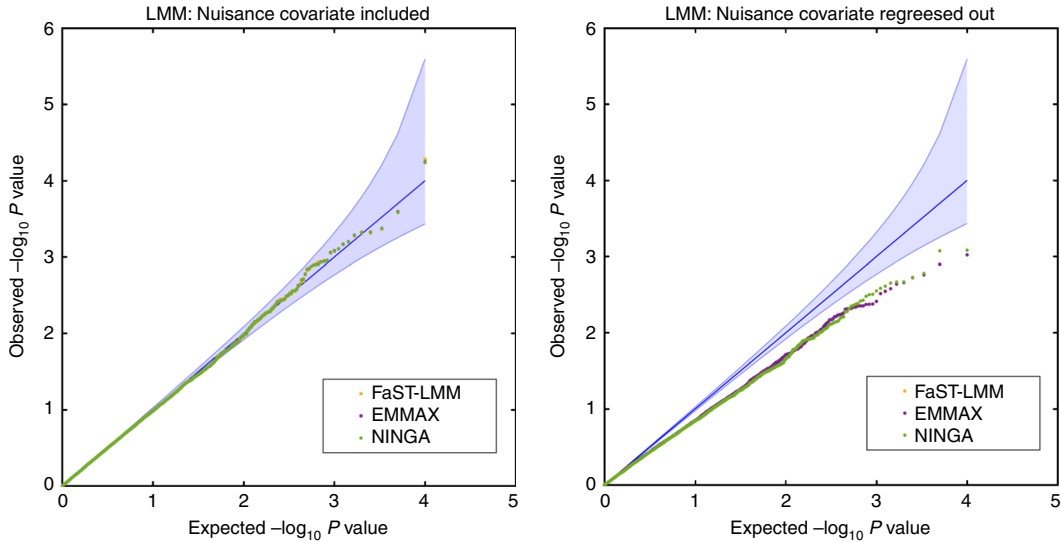

**Fig. 4** Comparison of parametric *P*-values from different software packages for controlling heritable nuisance covariate in GWA (Simulation 5). Results shown for the distribution of null ($\beta_{\text{SNP}} = 0$ and $\beta_{\text{nuisance}} = 0$). *P*-values from score statistic based the simplified REML function using non-iterative random effect estimator (NINGA), FaST-LMM and EMMAX, for the GRM from 4000 unrelated individuals with 5000 realizations when nuisance covariate is included in the LMM (left panel) and residualised phenotypes are fitted to LMM (right panel). The nuisance covariate heritability is 60%, due to 10 SNPs[25] using the additive model. There is no apparent differences between the packages when the nuisance covariate is included in the LMM. However, fitting the LMM to the residualised phenotypes produces invalid results in this setting of correlation between SNP and nuisance covariates. Confidence bounds created with the results of ref. [59], where ordered *P*-values follow beta distribution (see the Supplementary Note 1 for more details on simulation settings)

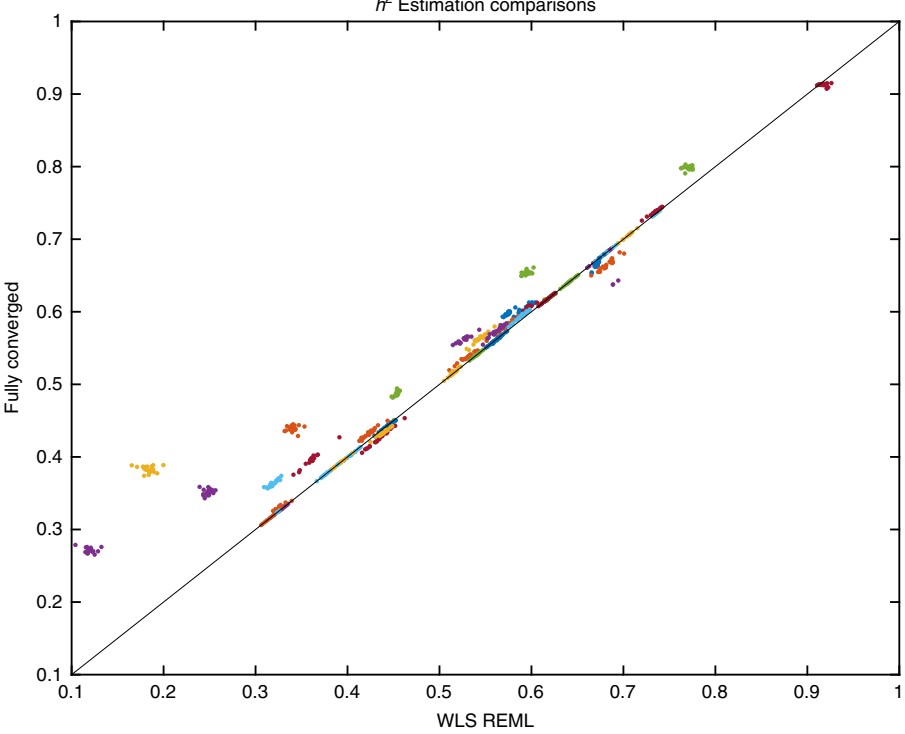

**Fig. 5** Comparison of one-step and fully converged random effect estimators of $\sigma_A^2$ based on the simplified REML function using real data analysis. Colors represent random effect estimation at different regions for all 22 chromosomes. The scatter plot show consistent trend towards underestimation of random effect using non-iterative method, though this apparent increased accuracy comes with a $10^9$-fold greater computation time

phenotypes require fitting a model at each point (voxel/element) in the brain, and the large number of measurements presents a challenge both in terms of computational intensity and the need to account for elevated false positive risk because of the multiple

testing problems both in terms of number of elements in image and number of markers being tested.

There has been rapid advances in quantitative genetic statistical methods account for population structure in GWA studies of

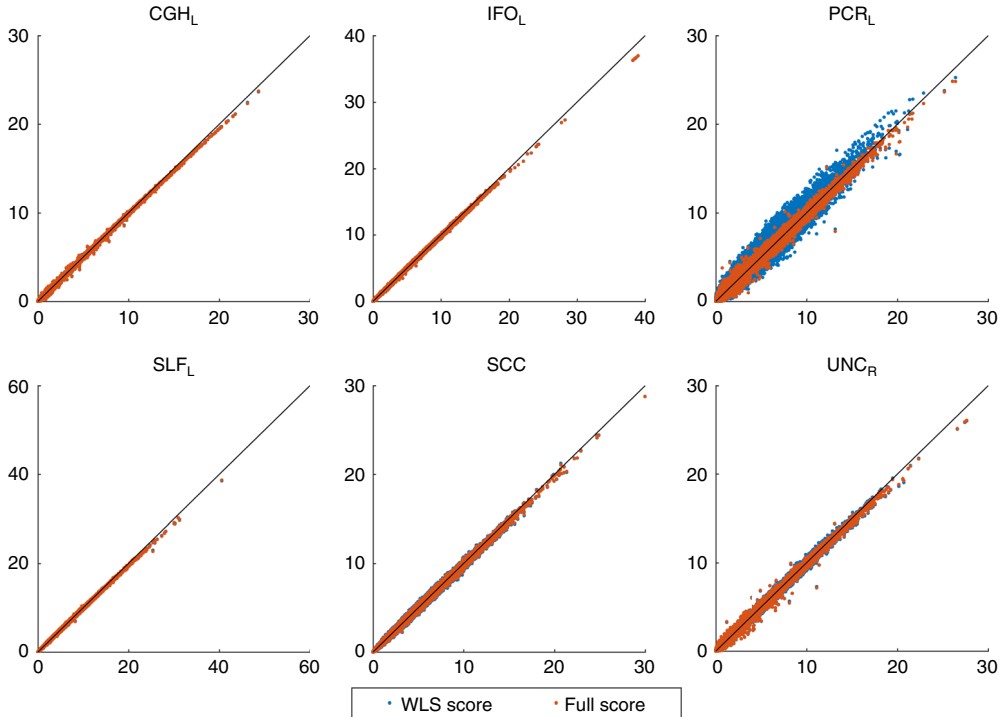

**Fig. 6** Comparison of test statistic values for association testing ($H_0:\beta_1 = 0$) using real data analysis. Results shown for the non-iterative and fully converged random effect estimators based on the simplified REML function and FaST-LMM's LRT. Each plot represents a ROI where x-axis shows FaST-LMM's LRT and y-axis represents the score test. Despite strong concordance between the score test results using WLS or fully converged random effect estimator, FaST-LMM is slightly more powerful, consistently for whole brain parcellation (see Supplementary Figures 10 and 11 for the rest of the ROIs)

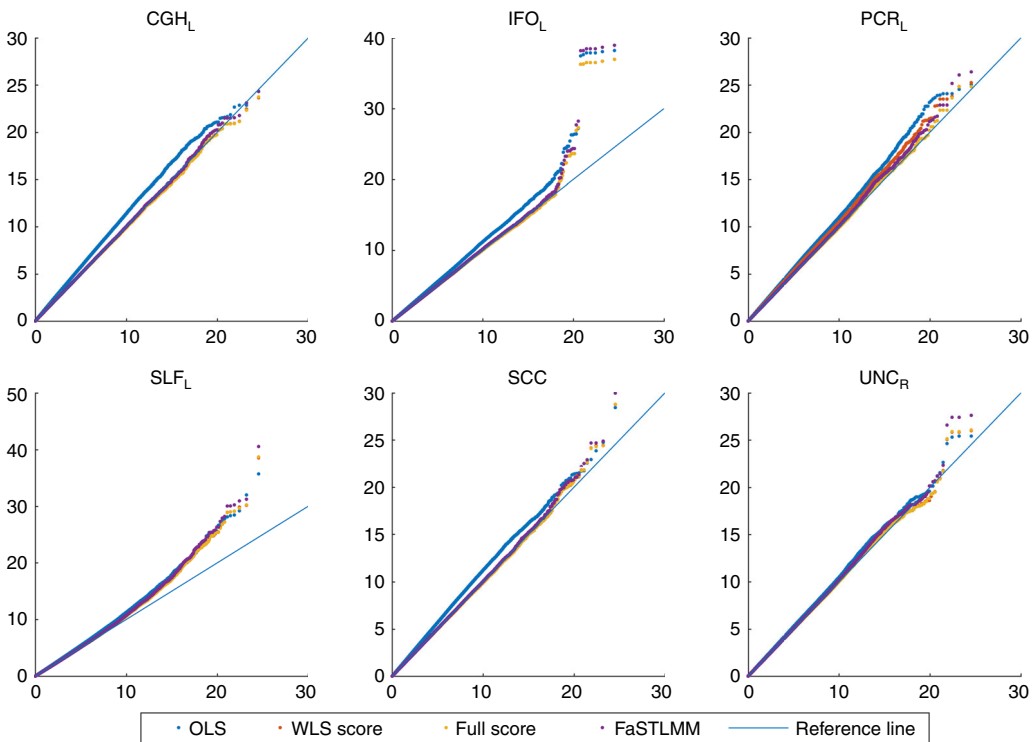

**Fig. 7** QQ plot for comparing FaST-LMM and the score test based on the simplified REML function using the WLS-REML random effect estimator with the linear regression with MDS as nuisance fixed effects (Real data analysis). Each plot corresponds to different ROIs. These plots show either an identical distribution or slightly larger values for the OLS approach. However, the OLS approach has poor genomic control (Supplementary Figure 9)

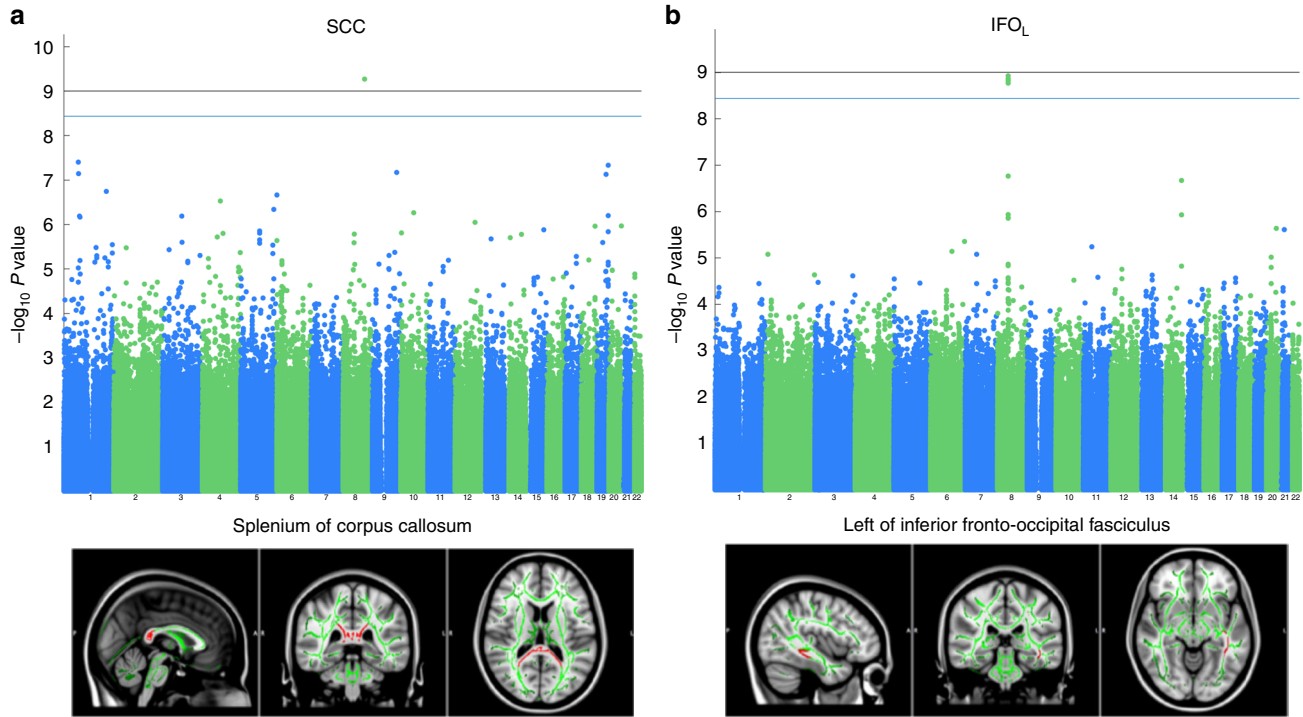

**Fig. 8** GWA of whole brain fractional anisotropy data, using a whole brain parcellation of 42 regions (real data analysis). Permutation test was used to derive FWE corrected P-values of score test based on the simplified REML function using one-step random effect estimator. Among the 42 × 1,376,877 ≈ 57 million statistics, eight passed the permutation based FWE threshold ($\chi_1^2 = 34.72$, blue line in Manhattan plot). Application of a Bonferroni correction for 42 tests to the usual GWA alpha level ($5 \times 10^{-8}$) yields to a more stringent threshold ($\chi_1^2 = 36.98$, black line in Manhattan plot) where only one association survives, indicating the potential improved power from a permutation-based inference that accounts for dependency among the tests

unrelated individuals. Linear mixed models allowing for the rigorous testing of genetic associations (and, more generally, any fixed effects) have long been employed in human genetics as the standard to correct for the non-independence among subjects due to known familial relatedness in pedigree-based studies[39–43]. The LMM has gained popularity as an alternate method for GWA of unrelated individuals to correct for population structure[15–26]. However, due to the required inversion of potentially large matrices, the general LMM is computationally intensive where the complexity includes the deriving of the GRM, variance component parameter estimation, fixed effect estimation, and the calculation of the required association statistic for each marker grows with sample size and number of candidate markers for association testing.

To tackle these problems, we used an orthogonal transformation that substantially reduced LMM complexity for GWA. The equivalence between projected model and REML function helped us to reduce complexity of association testing. Specifically, the projection reduces the information matrix to a scalar that enables efficient vectorized implementation of score test with time complexity $O(SNV)$. Further improvements in speed can be achieved by using the WLS-REML random effect estimator with $O(NV)$ that we found to be more accurate than the WLS-ML estimator.

We conducted intensive simulation studies, evaluating a broad set of test statistics for association testing using the simplified ML and REML functions accompanied by one-step and fully converged random effect estimators. The one-step random effect estimator using simplified REML function provides more accurate approximation of the fully converged one in comparison to the WLS-ML variance component estimator. The simulation and real data analysis shows that only minor differences in marker effect estimation and association test statistics between one-step and fully converged random effect estimator. However, the

former requires less computational resources. Also, we could not observe any appreciable differences in performances in terms of the error rate and power using the GRM from unrelated individuals or kinship matrix from a family study.

The WLS-REML random effect estimator is fast enough to be used to estimate voxel-wise heritability. Although the proposed one-step random effect estimator is not as accurate as fully converged one, it can be used for filtering a small number of elements for further investigation with more computational intense tools. Furthermore, when restricted to individuals with European ancestry we found LMM had genomic controls values closer to 1 than OLS values, indicating the success of the LMM in dealing with population structure.

We selected the score test based on the simplified REML function for further investigations because it only requires a single variance component estimate, common to all markers under the null hypothesis. Furthermore, efficient vectorized implementation of score test for images accelerates association testing. The null distribution of WLS-REML score test P-values was nearly as accurate as for the fully converged REML score test, meaning that permutation is not required for element-wise inference.

Genome wide association analysis of high dimensional imaging phenotype requires a computationally efficient LMM that can accommodate large sample sizes, provide multiple testing correction and complex spatial dependence among image elements. Although we have not observed significant differences of parametric P-values between our proposed method and existing methods, current LMM approaches depend on numerical optimization and are not feasible for GWA of over 1000's imaging phenotypes, such as found in the UKBiobank[44] or Human connectome project which has related individuals. Moreover, a search for genetic association across the genome with imaging

phenotypes requires intense multiple testing corrections both for number of elements in an image and number of markers. The reduced complexity of our proposed method due to a non-iterative random effect estimator and vectorized implementation of score statistic computation makes permutation-based FWE-corrected inferences feasible. Our permutation approach is more computationally efficient than others proposed in the genetics literature[45], avoiding inversion of the phenotypic covariance on each permutation. Permutation provides flexible FWE inference for ROIs or the whole brain, voxel-wise or cluster-wise. Though methods based on effective number of independent tests are computationally efficient, they depend on an (1) arbitrary GWAS threshold that depends on the chip used, (2) these approaches are not applicable to cluster-wise inference and (3) do not scale to voxel-wise inference which are the standard spatial statistics in imaging. While our vectorized implementation of score statistics demands complete data, there are various imputation methods can be used prior to association analysis to address missingness.

Whether using the linear mixed model for controlling population structure or kinship, high dimensional imaging phenotypes presents challenges in terms of computational intensity and elevated false positive risk; growing sample sizes and whole genome sequence data add to the computational burden. Our contribution in the acceleration of the exact LMM can be seen at two steps. First, covariance matrix estimation using WLS-REML random effect estimator reduces time complexity from $O(N^3 + INV)$ to $O(N^3 + NV)$. Further improvement in speed is also obtained by using the vectorized implementation of the score test based on simplified REML function. Our proposed method allows efficient implementation that reduces running time complexity to $O(SNV)$. In addition, the efficient score test computation is fast enough to allow the permutation test to control family-wise error rate for number of elements in image and number of markers, and allow the use of spatial statistics like cluster size or TFCE.

## Methods

**Reference methods.** Several approximate or exact methods have been proposed to speed up LMM-based testing. Approximate methods assume the total polygenic random effect is same for all markers under the null hypothesis of no marker effect, hence the relevant residual genetic variance component is estimated only once using all markers. In contrast, exact methods, which is the recommended LMM practice[19,20,25,35,46], estimate a residual variance component conditional on each marker's effect. In studies of "unrelated" subjects, this residual variance component often involves re-estimation of the GRM which is constructed excluding the candidate marker and surrounding markers in linkage disequilibrium.

To the best of our knowledge, the fastest implementation of exact LMM is Fast-LMM, which transforms the phenotype and LMM model with the genetic similarity matrix (GRM) eigenvectors and uses a profile likelihood approach to simplify variance component estimation. The eigenvector matrix diagonalisation along with the profile likelihood with only one variance parameter reduces optimization time substantially. In Fast-LMM the covariance matrix is estimated only under the null hypothesis of no marker effect, and then a generalized least squares (GLS) is applied to estimate the marker effect and the LRT is used for hypothesis testing. Note that small sample size behavior of this approach has not been validated; for example, using it for association analysis of imaging phenotypes with only 300 subjects might not be valid. In addition to concerns about the finite sample validity, Fast-LMM requires numerical optimization for each element (voxel/ROI) of image that makes it computationally intensive or essentially impractical for large-scale imaging phenotypes.

**Linear mixed effect models.** At each voxel/element, a LMM for the genetic association for $N$ individuals can be written as:

$$Y = X_1\beta_1 + X_2\beta_2 + g + \varepsilon, \tag{1}$$

where $Y$ is the $N$-vector of the measured phenotype; $X_1$ is a $N$-vector of a given marker's minor allele count, implementing an additive genetic model; $X_2$ is the $N \times (P-1)$ matrix containing an intercept and fixed effect nuisance variables like age and sex; $\beta_1$ is the scalar genetic effect; $\beta_2$ is the $(P-1)$-vector of nuisance regression coefficients and $g$ is the $N$-vector of latent (unobserved) additive genetic effects; and $\varepsilon$ is the $N$-vector of residual errors. The trait covariance, $\text{var}(Y) = \text{var}$

$(g + \varepsilon) = \Sigma$ can be written

$$\Sigma = \sigma_A^2(2\Phi) + \sigma_E^2 I, \tag{2}$$

where $\sigma_A^2$ and $\sigma_E^2$ are the additive genetic and the environmental variance components, respectively; $I$ is the identity matrix; and $2\Phi$ is the GRM matrix where element $(i, j)$ is calculated as:

$$\phi_{i,j} = \frac{1}{M}\sum_{k=1}^{M}\frac{(x_{ik} - 2p_k)(x_{jk} - 2p_k)}{2p_k(1 - p_k)},$$

where $x_{ik}$ is the minor allele count of the $i$th subject's $k$th marker, coded as coded as 0, 1 or 2; $p_k$ is frequency of the $k$th marker; and $M$ is the total number of markers.

Under the assumption that the the data follows a multivariate normal distribution, the model specified by Eqs. (1) and (2) have a log-likelihood of

$$\ell_{\text{ML}}(\beta_{\text{ML}}, \Sigma_{\text{ML}}; Y, X)$$
$$= -\frac{1}{2}[\text{Const} + \log(|\Sigma|) + (Y - X\beta)'\Sigma^{-1}(Y - X\beta)], \tag{3}$$

and a REML function of

$$\ell_{\text{REML}}(\Sigma_{\text{REML}}; Y, X)$$
$$= -\frac{1}{2}(\text{Const} - \log|X'X| + \log|\Sigma| + \log|X'\Sigma^{-1}X| + Y'PY), \tag{4}$$

where $X = \begin{bmatrix} X_1 & X_2 \end{bmatrix}$ and $\beta = \begin{bmatrix} \beta_1 & \beta_2 \end{bmatrix}$ are the full design matrix of fixed effects and their parameter estimate vector, respectively, and $P = \Sigma^{-1}(I - X(X'\Sigma^{-1}X)^{-1}X'\Sigma^{-1})$, the projection matrix. The fixed effect parameters are estimated using GLS

$$\hat{\beta}_{\text{REML}} = (X'\hat{\Sigma}_{\text{REML}}^{-1}X)^{-1}X'\hat{\Sigma}_{\text{REML}}^{-1}Y,$$

where $\hat{\Sigma}_{\text{REML}}^{-1}$ comes from optimized REML function (Eq. (4)).

Several algorithms have been proposed to accelerate ML or REML optimization by transforming the model with the eigenvectors of the GRM and/or using a different covariance matrix parametrisation[16,19,21,34,35,46]. Here we consider standard additive model covariance matrix parametrisation (Eq. (2)) as we can efficiently estimate it with our one-step, regression based approach[34].

**Simplified REML and ML functions.** The simplified ML function for LMM is discussed in refs. [16,19,21]. For completeness, we review shortly the simplified ML function, to be next followed by development of the simplified REML function. The simplified ML function is obtained by transforming the data and model with an orthogonal transformation $S$, the matrix of eigenvectors of $2\Phi$ that crucially coincide with the eigenvectors of $\Sigma$:

$$S'Y = S'X\beta + S'g + S'\varepsilon$$

which we write as

$$Y^* = X^*\beta + g^* + \varepsilon^*, \tag{5}$$

where $Y^*$ is the transformed data, $X^*$ is the transformed covariate matrix, $g^*$ and $\varepsilon^*$ are the transformed random components. The diagonalising property of the eigenvectors then gives a simplified form for the variance:

$$\text{var}(\varepsilon^*) = \Sigma^* = \sigma_A^2 D_g + \sigma_E^2 I,$$

where, $\Sigma^*$ is the variance of the transformed data and $D_g = \text{diag}\{\lambda_{gi}\}$ is a diagonal matrix of the eigenvalues of $2\Phi$.

The log likelihood takes on the exact same form as Eq. (3) for $Y^*, X^*, \beta$ and $\Sigma^*$, except is easier to work with since $\Sigma^*$ is diagonal:

$$\ell_{\text{ML}}(\beta, \sigma_A, \sigma_E; Y^*, X^*)$$
$$= -\frac{1}{2}\left[N\log(2\pi) + \sum_{i=1}^{N}\log(\sigma_A^2\lambda_{gi} + \sigma_E^2) + \sum_{i=1}^{N}\frac{(y_i^* - x_i^*\beta)^2}{\sigma_A^2\lambda_{gi} + \sigma_E^2}\right],$$

where $y_i^*$ is the $i$th element of $Y^*$, and $x_i^*$ is the $i$th row of $X^*$.

The REML function (Eq. (4)) is simplified using projection matrix ($P = \Sigma^{-1}(I - X(X'\Sigma^{-1}X)^{-1}X'\Sigma^{-1})$[35]). Also, an alternate simplified form of the REML function can be obtained as follows:

Let $M = I - X(X'X)^{-1}X'$ be the residual forming matrix based on the fixed effects regressors. Since $M$ is idempotent, it can be decomposed as

$$M = AA',$$
$$A'A = I,$$

where $A$ is the $N \times (N - P)$ matrix of the eigenvectors of $M$ corresponding to the non-zero eigenvalues. Crucially, $A$ also residualises the data, because it is

orthogonal to the design matrix $X$:

$$A'X = A'AA'X,$$
$$= A'MX = 0.$$

Hence $A'Y \sim N(0, A'\Sigma A)$ and the log likelihood of the transformed data is

$$\ell(A'Y, \Sigma)$$
$$= -\tfrac{1}{2}\left[\text{Const} + \log|A'\Sigma A| + Y'A(A'\Sigma A)^{-1}A'Y\right]. \quad (6)$$

Now we show that this (Eq. (6)) is equivalent to Eq. (4), and thus we can use the eigenvectors of the residual-forming matrix to build the REML log likelihood.

It can be shown that

$$\log|A'\Sigma A| = \log|\Sigma| + \log|X'\Sigma^{-1}X| - \log|X'X|.$$

and using $A(A'\Sigma A)^{-1}A' = P$ ([47], M4.f p.451), we have that Eqs. (4) and (6) are equivalent and the transformed data likelihood function is exactly as same as the REML function.

As $A$ is not unique, we seek to find one that diagonalises the covariance of the residualised data. The transformation matrix could be derived from eigendecomposition of GRM adjusted for the fixed effect covariates as follows:

$$M(2\Phi)M = S_r D_{g_r} S_r',$$

where $D_{g_r} = \text{diag}\{\lambda_{g,i}\}$ is the $(N-P) \times (N-P)$ diagonal matrix of non-zero eigenvalues; and $S_r$ is the $N \times (N-P)$ matrix of eigenvectors that corresponds to non-zero eigenvalues. Firstly, $S_r$ is a valid $A$, because its columns are orthogonal $S_r'S_r = I$ and $M = S_r S_r'$[16]. Thus we define the projected polygenic model by pre-multiplying $S_r'$ both sides of polygenic model (Eq. (1)):

$$S_r'Y = S_r'X + S_r'g + S_r'\varepsilon,$$
$$S_r'MY = S_r'MX + S_r'Mg + S_r'M\varepsilon.$$

which we write as

$$Y_r^* = g_r^* + \varepsilon_r^*, \quad (7)$$

where $Y_r^*$, $g_r^*$ and $\varepsilon_r^*$ are $N-P$ projected phenotype, genetic and residual vectors, respectively. In this fashion, the projected phenotype covariance matrix becomes diagonal:

$$\text{cov}(Y_r^*) = \Sigma_r^*,$$
$$= \text{cov}(S_r'Y),$$
$$= \text{cov}(S_r'MY),$$
$$= S_r'(M\Sigma M)S_r,$$
$$= S_r'(\sigma_A^2 M(2\Phi) + \sigma_E^2 M)S_r,$$
$$= \sigma_A^2 S_r'(S_r D_{g_r} S_r')S_r + \sigma_E^2 S_r'(S_r S_r')S_r,$$
$$= \sigma_A^2 D_{g_r} + \sigma_E^2 I,$$

where we have used the identity $S_r'M = S_r'$. That is, therefore the projected data, $Y_r^*$, loglikelihood takes on a simpler form:

$$\ell_{\text{REML}}(\sigma_A^2, \sigma_E^2; Y_r^*)$$
$$= -\tfrac{1}{2}\left[\text{Const} + \sum_{i=1}^{N-P}\log(\lambda_{g,i}\sigma_A^2 + \sigma_E^2) + \sum_{i=1}^{N-P}\frac{y_{ri}^{*2}}{\lambda_{g,i}\sigma_A^2 + \sigma_E^2}\right]. \quad (8)$$

where $y_{ri}^{*2}$ is the square of the $i$th element of $Y_r^*$. It is clear from the Eq. (8) that working with the simplified version of REML is computationally easier than the original one (Eq. (4)). Beside accelerating the REML optimisation, this approach facilitates performing LRT for fixed effects ($\beta$s) and leads to a computationally efficient estimator and test statistic, described below.

**REML and ML parameter estimation**. We choose Fisher's scoring method to optimize the simplified ML and REML functions because it leads to computationally efficient variance component estimators. The score and the expected Fisher information matrices for the simplified models can be expressed as:

$$S_{\text{ML}}(\beta, \theta) = \begin{bmatrix} X^{*'}\Sigma^{*-1}\varepsilon^* \\ -\tfrac{1}{2}[U'\Sigma^{*-1}1 - U'\Sigma^{*-2}\varepsilon^{*2}] \end{bmatrix},$$

$$I_{\text{ML}}(\beta, \theta) = \begin{bmatrix} X^{*'}\Sigma^{*-1}X^* & 0 \\ 0 & \tfrac{1}{2}U'\Sigma^{*-2}U \end{bmatrix},$$

and

$$S_{\text{REML}}(\theta) = -\tfrac{1}{2}\left[U_r'\Sigma_r^{*-1}1 - U_r'\Sigma_r^{*-2}Y_r^{*2}\right],$$
$$I_{\text{REML}}(\theta) = \tfrac{1}{2}U_r'\Sigma_r^{*-2}U_r,$$

where $\theta = (\sigma_E^2, \sigma_A^2)$; $U = [\mathbf{1}, \lambda_g]$ and $U_r = [\mathbf{1}_r, \lambda_g]$ are $N \times 2$ and $(N-P) \times 2$ matrices; and $\lambda_g$ is the vector of eigenvalues of $(2\Phi)$; $\lambda_{g_r}$ the vector of eigenvalues of $M(2\Phi)M$; $\mathbf{1}$ and $\mathbf{1}_r$ are $N$ and $(N-P)$-vectors of one, respectively; $Y_r^{*2}$ is the element wise square of $Y_r^*$; and $\varepsilon^{*2}$ is the element wise square of $\varepsilon^*$. Following Fisher's scoring method it can be shown that at each iteration, maximum-likelihood estimation of $\beta$ and $\theta$ are updated based on WLS regression of $Y^*$ on $X^*$ and $\varepsilon^{*2}$ on $U$, respectively, as follows:

$$\hat\beta_{\text{ML},j+1} = \left(X^{*'}(\hat\Sigma_j^*)^{-1}X^*\right)^{-1}X^{*'}(\hat\Sigma_j^*)^{-1}Y^*,$$
$$\hat\theta_{\text{ML},j+1} = \max\left\{0, \left(U'(\hat\Sigma_j^{*2})^{-1}U\right)^{-1}U'(\hat\Sigma_j^{*2})^{-1}\hat\varepsilon_j^{*2}\right\}, \quad (9)$$

and REML estimation of $\theta$ is updated based on WLS regression of $Y_r^{*2}$ on $U_r$ as follows:

$$\hat\theta_{\text{REML},j+1} = \max\left\{0, \left(U_r'(\hat\Sigma_{rj}^{*2})^{-1}U_r\right)^{-1}U_r'(\hat\Sigma_{rj}^{*2})^{-1}Y_r^{*2}\right\},$$

where $j$ indexes iteration; $\Sigma_j^{*2}$ and $\Sigma_{rj}^{*2}$ are constructed with $\theta_{\text{ML},j}$ and $\theta_{\text{REML},j}$, respectively; $\varepsilon_j^{*2}$ is the element-wise square of $\varepsilon_j^* = Y^* - X^*\beta_{\text{ML},j}$; $Y_r^{*2}$ is the element-wise square of $Y_r^*$; and the variance parameters $\theta$ must be positive, hence the maximum operator. As usual, these updates are iterated until convergence criteria holds.

It has been shown that when the initial value is a consistent estimator, the estimator based on the first iteration is asymptotically normal and consistent[48]. Such initial value for $\hat\beta_{\text{ML}}$ and $\hat\theta_{\text{ML}}$ could be derived from OLS regression coefficients of $Y^*$ on $X^*$ and squared residuals on $U$, respectively:

$$\hat\beta_{\text{ML,OLS}} = (X^{*'}X^*)^{-1}X^{*'}Y^*,$$
$$\hat\theta_{\text{ML,OLS}} = \max\{0, (U'U)^{-1}U'\varepsilon^{*2}\}.$$

For REML, initial values for $\hat\theta_{\text{REML,OLS}}$ can be found as OLS regression coefficient of $Y_r^{*2}$ on $U_r$:

$$\hat\theta_{\text{REML,OLS}} = \max\{0, (U_r'U_r)^{-1}U_r'Y_r^{*2}\}.$$

Hence our one-step, non-iterative estimators are:

$$\hat\beta_{\text{ML,WLS}} = \left(X^{*'}(\hat\Sigma_{\text{OLS}}^*)^{-1}X^*\right)^{-1}X^{*'}(\hat\Sigma_{\text{OLS}}^*)^{-1}Y^*, \quad (10)$$

$$\hat\theta_{\text{ML,WLS}} = \max\left\{0, \left(U'(\hat\Sigma_{\text{OLS}}^{*2})^{-1}U\right)^{-1}U'(\hat\Sigma_{\text{OLS}}^{*2})^{-1}\varepsilon_{\text{OLS}}^{*2}\right\}, \quad (11)$$

$$\hat\theta_{\text{REML,WLS}} = \max\left\{0, \left(U_r'(\hat\Sigma_{\text{OLS,r}}^{*2})^{-1}U_r\right)^{-1}U_r'(\hat\Sigma_{\text{OLS,r}}^{*2})^{-1}Y_r^{*2}\right\}, \quad (12)$$

where $\hat\Sigma_{\text{OLS}}^*$ and $\hat\Sigma_{\text{OLS,r}}^*$ are formed by $\hat\theta_{\text{ML,OLS}}$ and $\hat\theta_{\text{REML,OLS}}$ respectively, and $\hat\varepsilon_{\text{OLS}}^{*2}$ is the element-wise square of $\hat\varepsilon_{\text{OLS}}^* = Y^* - X^*\hat\beta_{\text{OLS}}$. Going forward, we will use "ML" or "REML" to refer to the iterated estimators and "WLS" to refer to these one-step estimators.

**Association testing**. The score, LRT and the Wald tests can be used for the genetic association testing using either ML or REML functions of the model in Eq. (1).

The score statistic[49] that requires the value of score and information matrices under the the null hypothesis constraint ($H_0: \beta_1 = 0$) for the simplified ML model (Eq. 5) can be written

$$T_{\text{S,ML}} = \tilde\varepsilon_{\text{ML}}'\tilde\Sigma_{\text{ML}}^{*-1}X_1^*\left[C'(X^{*'}\tilde\Sigma_{\text{ML}}^{*-1}X^*)^{-1}C\right]X_1^{*'}\tilde\Sigma_{\text{ML}}^{*-1}\tilde\varepsilon_{\text{ML}},$$

where $C$ is a $P \times 1$ contrast vector; $X^* = [X_1^* \quad X_2^*]$ encompasses the full transformed covariate matrix; $\tilde\varepsilon_{\text{ML}}'$ and $\tilde\Sigma_{\text{ML}}$ are the ML residual and covariance matrix estimates under the null hypothesis constraint. The score statistic for the projected model (Eq. (7)) can be derived like $T_{\text{S,ML}}$ following the projection with respect to the $H_0$ fixed effects, i.e. nuisance, terms $X_2$:

$$T_{\text{S,REML}} = Y_r^{*'}\tilde\Sigma_{1r}^{*-1}X_{1r}^*\left(X_{1r}^{*'}\tilde\Sigma_{1r}^{*-1}X_{1r}^*\right)^{-1}X_{1r}^{*'}\tilde\Sigma_{1r}^{*-1}Y_r^*, \quad (13)$$

where $Y_r^* = S_{2r}'Y$ and $X_{1r}^* = S_{2r}'X_1$ are $(N-P+1)$-vectors of the projected phenotype and allele frequency count, respectively; and the projection matrix $S_{2r}$ is

comprised of the eigenvectors of $M_2(2\Phi)M_2$ with non-zero eigenvalues, $M_2 = I - X_2(X_2X_2')^{-1}X_2'$; and $\tilde{\Sigma}_r^{*-1}$ is the projected model covariance matrix estimation under the null model constraint.

The LRT[50] statistic is twice the difference of the optimized log-likelihoods, unrestricted minus $H_0$-restricted. For ML this requires optimizing the likelihood function twice, once under the null $H_0:\beta_1 = 0$, once under the alternative. We denote the test statistic for this test $T_{L,ML}$. A well-known aspect of REML is that it cannot be used to tests of fixed effects, since the null hypothesis would represent a change of the projection that defines the REML model. However, we can consistently use the same projection $S_{2r}$, under the unrestricted and restricted models, to diagonalise our covariance and carry out such a hypothesis test. To be precise, the unrestricted model is

$$S_{2r}'Y = S_{2r}'X_1\beta_1 + S_{2r}'g + S_{2r}'\varepsilon,$$

where $S_{2r}'X_2\beta_2 = 0$ by the construction of $S_{2r}$, and the restricted model is

$$S_{2r}'Y = S_{2r}'g + S_{2r}'\varepsilon.$$

Following the same procedure as ML, the test statistic is denoted by $T_{L,REML}$.

For a scalar parameter, the Wald test[49] is the parameter estimate divided by the standard deviation of the estimate under an unrestricted model. For an vector parameter $\beta$ and contrast $C$, it takes the form

$$T_W = C\hat{\beta}\big(C(X'\hat{\Sigma}^{-1}X)C'\big)^{-1}\hat{\beta}'C'$$

where $\hat{\beta}$ and $\hat{\Sigma}^{-1}$ are the parameter estimations under the alternative hypothesis; this form holds for both ML and REML. A test for genetic association testing can be calculated either using fully converged or one-step variance component estimators. In the parametric framework, all of the aforementioned tests null distribution follow chi square distribution with one degree of freedom asymptotically.

**Inference using the permutation test**. In neuroimaging the permutation test is a standard tool to conduct inference while controlling the family wise error rate (FWE)[33]. It only requires an assumption of exchangeability, that the joint distribution of the error is invariant to permutation, and provides exact inference in the absences of nuisance variables, or approximately exact inference with nuisance variables[51]. Control of the FWE of a voxel-wise or cluster-wise statistic is obtained from a maximum distribution of the corresponding statistic. However naive use of permutation test for genetic association testing, ignoring dependence structure between individuals, leads to invalid inferences[45,52]. To the best of our knowledge, the only work that uses a permutation test for association analysis in the context of LMM uses the estimated covariance matrix under the null hypothesis to whiten the phenotype vector, yielding exchangeable data[45]. This approach requires estimating the covariance matrix for each phenotype for each permutation, a significant computational burden with high dimensional imaging data. Here we propose two permutation schemes that account for dependence explained by our model, one free and one constrained permutation approach.

The genetic association testing in the context of LMM using a permutation test requires proper handling of fixed effect and random effect nuisance variables in order to respect the exchangeablity assumption. While there are a variety of methods for testing for a fixed effect when the errors are independent[51]. However, little work has been done for fixed effect inference using a permutation test in linear mixed models where the error term is correlated[45].

Free permutation for the simplified ML model: For the simplified model (Eq. (5)) we create permuted data $\tilde{Y}^*$ using the reduced, $H_0:\beta_1 = 0$ null model residuals to create surrogate null data,

$$\tilde{Y}^* = X_2^*\hat{\beta}_2 + \tilde{P}\hat{\varepsilon},$$

where $\tilde{P}$ is one of $N!$ possible $N \times N$ permutation matrices; $\hat{\beta}_2$ is the reduced model nuisance estimate found with either fully converged (Eq. (9)) or one-step (Eq (10)) methods; $\hat{\varepsilon}$ denotes the reduced model residuals likewise found with either fully converged or one-step estimators; and the tilde accent on the data $(\tilde{Y}^*)$ and permutation $(\tilde{P})$ denotes one of many null hypothesis realizations. The reduced null model is not exchangeable due to heteroscedasticity of $\Sigma^*$, but we account for this in each permutation step by fitting the simplified model (5) with the permuted covariance matrix

$$\text{cov}(\tilde{Y}^*) = \tilde{P}\Sigma^*\tilde{P}' = \sigma_A^2\tilde{P}D_{g}\tilde{P}' + \sigma_E^2I.$$

With this approach we obtain samples from the empirical null distribution of the maximum score, LRT and the Wald tests (or cluster-size, after thresholding one of these test statistics), where the maximum is taken over all voxels and SNPs to control FWE.

Free permutation for simplified REML model: While the previous approach creates null hypothesis realizations by permuting the null-model residuals and adding back on estimated nuisance, here we will permute data after reduced-model eigen-transformation. We do this because projection removes the nuisance fixed

effect covariates. In both cases, though, we must account for the dependence existing under $H_0$.

An alternate permutation scheme could be built based on projecting the LMM model (Eq. (1)) to the lower dimension space with respect to the null hypothesis reduced model, i.e. using only the nuisance fixed effect terms. Let $M_2$ be the residual forming based on $X_2$ alone and define $S_{2r}$ as the transformation based on the non-trivial eigenvectors of $M_2(2\Phi)M_2$, creating a model with dimension $N - (P+1)$:

$$Y_r^* = X_{1r}^*\beta_1 + g_r^* + \varepsilon_r^*,$$

where $Y_r^* = S_{2r}'Y$ is the reduced transformed data; $X_{1r}^* = S_{2r}'X_1$ is as above, the reduced transformed additive genetic effect; $g_r^* = S_{2r}'g$ and $\varepsilon_r^* = S_{2r}'\varepsilon$ are the latent genetic effect and random error terms, respectively, after the reduced transformation. Here we permute the data, producing $\tilde{Y}_r^* = \tilde{P}Y_r^*$, with permuted covariance matrix

$$\text{cov}(\tilde{Y}_r^*) = \sigma_A^2\tilde{P}D_{g_r}\tilde{P}' + \sigma_E^2I$$

fit in each permutation step. However, under the null hypothesis of no genetic effect, estimated random effects for permuted phenotype are exactly as same as the un-permuted phenotype and hence the variance components only need to be estimated once.

Constrained permutation for exchangeability blocks: In the free permutation approaches we permute despite the lack of exchangeability, but then permute the covariance structure to account for this. An alternate approach is to define exchangeability blocks where observations within each block can be regarded as exchangeable. Precisely, with exchangeability blocks, the assumption is that permutations altering the order of observations only within each block preserve the null hypothesis distribution of the full data.

While not feasible for the original correlated model (1), in the simplified ML (5) or simplified REML (7) model we can define approximate exchangeability blocks. In simplified models the sorted eigenvalues arrange the observations by variance (increasing or decreasing, depending on software conventions). Hence blocks of contiguous observations $Y^*$ or $Y_r^*$ will have variance that is as similar as possible and will be, under the null hypothesis, approximately exchangeable.

We propose to define exchangeability blocks such that the range of $D_g$ or $D_{g_r}$ values within a block is no greater than 0.01. This cut off ensured the eigenvalues did not vary by more than a factor of 10% within a block. Permutation is constrained within these blocks and the test procedure is as described above for simplified ML and REML free permutation schemes above, except that the test statistic is computed using the unpermuted covariance matrix.

**Efficient score statistic implementation for vectorized images**. To fully exploit the computational advantage of our non-iterative, reduced-dimension projected model estimation method we require a vectorized algorithm. That is, even without iteration, the method will be relatively slow if the evaluation of the estimates is so complex that each phenotype must be looped over one-by-one. For fast evaluation with a high-level language like Matlab, the estimation process for a set of phenotypes must be cast as a series of matrix algebra manipulations. To exploit the computational advantage of this approach, the phenotype should be observed for all individuals. However, soft imputation methods can be used in case of missing phenotypes. An alternate approach has been proposed[53] for multi-trait GWA. However, our proposed method is computationally more efficient due to NINGA.

In this section, we develop the vectorized algorithm for association one chromosome's worth of SNPs and all image voxels/elements (subject to memory constraints). To avoid proximal contamination[19] and efficient implementation of LMM, we follow leave one chromosome out approach where all markers on a chromosome being tested are excluded from the GRM[25,27].

Let $Y_r$ and $X_r$ be a $(N - P) \times V$ and $(N - P) \times G$ matrices of projected traits and allele frequencies, respectively, where $V$ and $G$ are number of elements in image and number of SNPs the tested chromosome, respectively. The score test requires parameter estimation under the null hypothesis constraints, and since $X_2$ is the same for all SNPs, the estimated covariance matrix will be the same all markers the chromosome. Thus the covariance matrix only need to be estimated once as follows:

$$\mathbf{F} = \mathbf{Y}_r \odot \mathbf{Y}_r,$$
$$\theta = \max((U_r'U_r)^{-1}U_r'\mathbf{F}, \mathbf{0}),$$
$$W = \mathbf{1}_{NV} \oslash ((U_r\theta) \odot (U_r\theta)),$$

where $\mathbf{F}$ and $\mathbf{Y}_r$ are $(N - P) \times V$ matrices, where each column of $\mathbf{Y}_r$ is $Y_r^*$ (Eq. (7)) for one image element and $\mathbf{F}$ is the element-wise squaring of $\mathbf{Y}_r$; $\odot$ denotes Hadamard product; $\oslash$ denotes element-wise division; $\theta$ is the $2 \times V$ matrix of OLS solutions which is matrix counterpart of $\hat{\theta}_{REML,OLS}$; $\mathbf{0}$ is the $2 \times V$ matrix of zeros; and here $\max(\cdot, \cdot)$ is an element-wise maximum between the two operands, evaluating to a $2 \times V$ matrix; $W$ and $\mathbf{1}_{NV}$ are the $(N-P) \times V$ matrices, where each column of $W$ is $\text{diag}\left(\hat{\Sigma}_{OLS,r}^{*-2}\right)$ for the corresponding image element and $\mathbf{1}_{NV}$ is a

matrix of ones. With the following notation,

$$A = \mathbf{1}_V W,$$
$$B = D'_g W,$$
$$C = \left(D'_g \odot \lambda'_g\right) W,$$
$$D = \mathbf{1}_V (W \odot F),$$
$$E = D'_g (W \odot F),$$

where $\mathbf{1}_V$ is the length-$V$ column vector of ones, we can compute the variance components of the vectorized image as

$$\boldsymbol{\sigma}^2_A = \max((-B \odot D + A \odot E) \oslash (A \odot C - B \odot B), \mathbf{0}), \quad (14)$$

$$\boldsymbol{\sigma}^2_E = \max((C \odot D - B \odot E) \oslash (A \odot C - B \odot B), \mathbf{0}), \quad (15)$$

$$\mathbf{S} = U_r \begin{bmatrix} \mathbf{1}_V \oslash \boldsymbol{\sigma}^2_E \\ \mathbf{1}_V \oslash \boldsymbol{\sigma}^2_A \end{bmatrix},$$

where $\boldsymbol{\sigma}^2_A$ and $\boldsymbol{\sigma}^2_E$ are the length-$V$ column vectors of genetic and environmental variance components, respectively; and $\mathbf{S}$ is a $(N - P) \times V$ matrix which here each column of $\mathbf{S}$ is the element-wise reciprocal of the diagonal of the variance matrix of the corresponding image element's data $\mathbf{Y}_r$ for each element of image. In this fashion, the score statistic matrix for all markers being tested and the vectorized image can be expressed as:

$$\mathbf{T}_{S,R} = [(\mathbf{X}'(\mathbf{S} \odot \mathbf{Y}_r)) \odot (\mathbf{X}'(\mathbf{S} \odot \mathbf{Y}_r))] \oslash [(\mathbf{X} \odot \mathbf{X})\mathbf{S}], \quad (16)$$

where $\mathbf{T}_{S,R}$ is a $G \times V$ matrix of score statistics for all SNPs and traits.

**Real data**. To validate our proposed methods for association estimation and inference for imaging data, we applied them on a dataset from healthy and schizophrenic individuals to perform ROI and voxel-wise genome wide association analysis using cluster wise inference. The sample was 54% healthy individual (175 control/155 schizophrenic) and had a mean age of 39.12 (SD = 14.9) where 50% of the sample is male[54].

**Diffusion tensor imaging**. Imaging data was collected using a Siemens 3T Allegra MRI (Erlangen, Germany) using a spin-echo, EPI sequence with a spatial resolution of $1.7 \times 1.7 \times 4.0$ mm. The sequence parameters were: TE/TR = 87/5000 ms, FOV = 200 mm, axial slice orientation with 35 slices and no gaps, 12 isotropically distributed diffusion weighted directions, two diffusion weighting values ($b = 0$ and 1000 s/mm$^2$), the entire protocol repeated three times.

ENIGMA-DTI protocols for extraction of tract-wise average FA values were used. These protocols are detailed elsewhere[55] and are available online http://enigma.ini.usc.edu/protocols/dti-protocols/. Briefly, FA images from subjects were non-linearly registered to the ENIGMA–DTI target brain using FSL's FNIRT[55]. This target was created as a "minimal de-formation target" based on images from the participating studies as previously described (Jahanshad et al., 2013b). The data were then processed using FSL's tract-based spatial statistics (TBSS; http://fsl.fmrib.ox.ac.uk/fsl/fslwiki/TBSS) analytic method[56] modified to project individual FA values on the hand-segmented ENIGMA-DTI skeleton mask. The protocol, target brain, ENIGMA–DTI skeleton mask, source code and executables, are all publicly available (http://enigma.ini.usc.edu/ongoing/dti-working-group/). The FA values are normalized across individuals by inverse Gaussian transform[57,58] to ensure normality assumption. Finally, we analyzed the voxel and cluster-wise FA values with applying along the ENIGMA skeleton mask.

**Genetic quality control**. In this study only genotyped single nucleotide polymorphisms (SNPs) from genome-wide information were included in the analysis. Visual inspection of multi-dimensional scaling analysis was used to extract individuals with European ancestry. SNPs from individuals with European ancestry that do not meet any of the following quality criteria were excluded: genotype call rate 95%, significant deviation from Hardy–Weinberg equilibrium $P < 10^{-6}$ and minor allele frequency 0.1 was used to ensure that sufficient numbers of subjects would be found in our sample in each genotypic group (homozygous major allele, heterozygous, homozygous minor allele) using an additive genetic model. After all quality control steps, 962,885 out of 1,406,990 SNPs remain for genome-wide association analysis.

**Data availability**. The data that support the findings of this study are available from Dr. Peter Kochunov upon reasonable request.

**Code availability**. Software implementation of this method, Nonparametric Inference for Genetics Analysis (NINGA), is available at http://nisox.org/software/ninga/.

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

## Acknowledgements

T.E.N. is supported by the Wellcome Trust. H.G, T.E.N. and P.K were supported by National Institutes of Health R01 EB015611-01.

## Author contributions

H.G. developed the statistical method, ran the simulations, analyzed the simulated and real data, and lead the writing of the manuscript. P.K. provided the human data and supervised its analysis. B.D. implemented the method both in CPU and GPU. A.M.W., D. C.G., J.B. and T.E.N. contributed to revisions of the paper. T.E.N. supervised the project. All authors contributed to critical review of the manuscript during its preparation.

## Additional information

**Competing interests:** The authors declare no competing interests.

