## [Peer Review File · Nature Communications]

Reviewer #1 (Remarks to the Author):

Ganjgahi et al. propose a number of tricks and improvements to linear mixed models to scale these up to large numbers of traits.

Individually, these are of minor novelty, but together they are relevant and useful.

The novel points are as follows:

1) The use of a single optimization step instead of a converged iterative estimator of the model to speed up computations.

2) Fast permutation tests based on transformed data.

3) A special purpose vectorized implementation of the LMM for the application of many phenotype studies. While these speedups are not of algorithmic nature, but rather a specialized implementation, they are still relevant, as they are needed for addressing the application of testing for associations with a large number of voxels in MRI images.

Detailed comments:

- Regarding 3)

In order for the speedups to work, all traits need to be observed for all individuals. As this assumption does not hold in many other multiple-phenotype scenarios, the assumption should be stated in the paper explicitly.

In this context, a related multiple phenotype LMM paper that makes the same assumption should be referenced: Fabregat-Traver and Bientinesi, ACM TOMS 2014: "Computing Petaflops over Terabytes of Data: The Case of Genome-Wide Association Studies"

- The derivations presented in the paper are based on a simplified REML formula. For this formula, the authors refer to Ganjgahi et al. 2015, but the identical formula also has been derived in Lippert et al, Bioinformatics 2014 "Greater Power..." also in a variance components testing context (See SI. Eq. 2).

Please add the corresponding reference.

In fact, the Python version of FaST-LMM implements the simplified REML for fixed effect testing with an LRT, integrating out nuisance fixed effects as in REML and performs ML estimation on the SNP effect that is being tested only.

Minor comments:

- Figure and Equation references in the Main to the Supplement are broken.

- The experiment and the content of the Figure in SI Figure 8 should be described better.

I hope you find these comments useful.

Reviewer #2 (Remarks to the Author):

The authors present a method for linear mixed modelling for neuroimaging phenotypes that scales well compared to existing approaches.

The authors claim two major contributions. First, they claim their use of score tests is novel, although the approach they use is very similar to their previous work (reference 40). The use of Wald, Likelihood Ratio and Score tests for a linear mixed model are implemented in the GEMMA software by Zhou and Stephens (see the software manual for this work).

The second claim of novelty that they reduce complexity by eigen-rotating the model is not true. The simplified ML function is not due to reference 40, or to reference 33 that is quoted in that paper. The use of the eigenvectors of the GRM to diagonalize the covariance structure of the model is due to references 24 and 26. This has been a standard method now for several years. The authors have not correctly credited this earlier work and have overstated the use of this approach as novel.

The authors use permutation to account for the correlation between the tested phenotypes. There are methods that can be used to estimate an effective number of tests (see for example

<https://www.ncbi.nlm.nih.gov/pmc/articles/PMC3325408/>) which could be applied here to estimate the correction for the 42 ROIs analysed. How does applying this method compare to using permutation?

A standard approach in LMM fitting has been to fit a model without any SNPs as fixed effects to estimate the variance components. These are then kept fixed when testing each SNP. This was first suggested in the EMMAX approach. Any covariates can be regressed out in advance so that a residual phenotype is used. I would like to see how the authors approach compares to this approach on the simulated and real datasets.

How does the permutation scheme used compare to that used in reference 46. If it is the same that should be acknowledged. If it is not, then a comparison should be included in the paper.

We thank the editor and the reviewers for their constructive comments. We have made significant revisions in response, marking our changes to the manuscript in blue.

Below we provide point-by-point responses, typesetting the reviewer/editor's comments in italics.

Editor's comment:

We hope you will find the referees' comments useful as you decide how to proceed. We would like to point out that for further consideration in Nature Communications, we would have to see more extensive benchmarking as requested by Reviewer #2 and demonstration of superiority (not just in speed but also accuracy) over these other methods.

In addition to our method being able to scale to the large scale phenotypes found with imaging, we have shown that we have more accurate standard errors (less biased) than FaST-LMM (FIGURE 8 Supplementary material) and can obtain exact, permutation-based familywise error (FWE) for spatial inferences (voxel-wise and cluster-wise), something no other method is able to provide.

Reviewer #1

1. *In order for the speedups to work, all traits need to be observed for all individuals. As this assumption does not hold in many other multiple-phenotype scenarios, the assumption should be stated in the paper explicitly.*

Response: We have addressed this point and mentioned the assumption explicitly in discussion (p8 last paragraph) and section 1.4 of online method (p16).

2. *In this context, a related multiple phenotype LMM paper that makes the same assumption should be referenced: Fabregat-Traver and Bientinesi, ACM TOMS 2014: "Computing Petaflops over Terabytes of Data: The Case of Genome-Wide Association Studies".*
3. *The derivations presented in the paper are based on a simplified REML formula. For this formula, the authors refer to Ganjgahi et al. 2015, but the identical formula also has been derived in Lippert et al, Bioinformatics 2014 "Greater Power..." also in a variance components testing context (See SI. Eq. 2).*

Response: We apologize for missing the references regarding 2 & 3. They are now added to the manuscript (p2 last paragraph, p10 last paragraph. P15 1st paragraph of section 1.5).

4. *Minor comments:*
 - *Figure and Equation references in the Main to the Supplement are broken.*
 - *The experiment and the content of the Figure in SI Figure 8 should be described better.*

Response: Thank you catching these; we have fixed the figures and equation referencing issue and elaborated more on simulation settings.

Reviewer #2

1. *The authors claim two major contributions. First, they claim their use of score tests is novel, although the approach they use is very similar to their previous work (reference 40). The use of Wald, Likelihood Ratio and Score tests for a linear mixed model are implemented in the GEMMA software by Zhou and Stephens (see the software manual for this work).*

The second claim of novelty that they reduce complexity by eigen-rotating the model is not true. The

simplified ML function is not due to reference 40, or to reference 33 that is quoted in that paper. The use of the eigenvectors of the GRM to diagonalize the covariance structure of the model is due to references 24 and 26. This has been a standard method now for several years. The authors have not correctly credited this earlier work and have overstated the use of this approach as novel.

Response: We apologize that we did not state original contributions more clearly. We have made changes to make our contribution clear (p2). Our proposed method reduces the complexity of LMM in GWA of high dimensional imaging data as follows:

- a. Use of non-iterative estimator for the random effects variance instead of fully converged estimator based on REML; and
- b. Vectorisation of score test computation, feasible due to the simplified REML function, vastly reducing the computational complexity of association testing.

The two in combination accelerate the model fitting process sufficiently to allow permutation inference, important for FWE control of spatial inference tools like the cluster size test.

To be clear, we did not intend to imply covariance matrix diagonalization or use of a score statistic as our major contribution. While, the non-iterative random effect estimator was introduced in our previous work, that was based on the simplified ML function, and here we have used the simplified REML function; in particular, we have shown that this improves the bias and mean square error of the estimated variance.

2. *The authors use permutation to account for the correlation between the tested phenotypes. There are methods that can be used to estimate an effective number of tests (see for example <https://www.ncbi.nlm.nih.gov/pmc/articles/PMC3325408/>) which could be applied here to estimate the correction for the 42 ROIs analysed. How does applying this method compare to using permutation?*

Response: We thank the reviewer for this reference. We have calculated a FWE threshold using the method proposed by the reviewer (p6). Application of a Bonferroni correction with effective number of tests yields a slightly more stringent threshold, but finds the same 8 association statistics as found with permutation. While this result would seem to undermine the need for permutation, we have noted that (1) this use of the effective number of independent tests depends on an arbitrary GWAS threshold that depends on the chip used, (2) the approach is not applicable to cluster-wise inference and (3) doesn't scale to voxel-wise inference which are the standard spatial statistics in imaging.

3. *A standard approach in LMM fitting has been to fit a model without any SNPs as fixed effects to estimate the variance components. These are then kept fixed when testing each SNP. This was first suggested in the EMMAX approach. Any covariates can be regressed out in advance so that a residual phenotype is used. I would like to see how the authors approach compares to this approach on the simulated and real datasets.*

Response: The reviewer raises a valid point, as a model based on precomputed residualised phenotypes would indeed save computation. However, we have found that this approach is not safe in general, and have added a simulation study to address this point (p6). First, we note that imaging association studies routinely use intracranial volume (ICV) as a nuisance covariate, and ICV is well known to have high heritability (h^2 ranges: 0.81-0.91, Peper et al 2007, 0.72-0.92, Glahn et al. 2007), hence we conducted a simulation study that evaluates controlling for a heritable fixed effect nuisance covariate in the null simulation setting when there is neither a SNP effect nor a covariate effect on the phenotype. Figure 4 compares performance of FaST-LMM, EMMAX and the score test based on the simplified REML function using non-iterative random effect estimator (NINGA). The simulation results show that the parametric P-value from all approaches when the nuisance covariate is included in the LMM is valid (Figure 4 left panel). However, the null distribution of P-values from LMM fitted on residualised phenotypes can be conservative (Figure 4 right panel).

4. *How does the permutation scheme used compare to that used in reference 46. If it is the same that should be acknowledged. It is not, then a comparison should be included in the paper.*

Response: We regret that we did not describe the permutation scheme in Abney (2015) and compare it to our proposed schemes. This is now addressed (p 14). The permutation scheme in Abney (2015) uses the estimated covariance matrix under the null hypothesis to whiten the phenotype vector to unit variance, yielding exchangeable data. This approach requires estimating the covariance matrix for each phenotype for each permutation, a significant computational burden with high dimensional imaging data. Although

our free permutation schemes based on the simplified ML or REML functions are close to this idea, our schemes do not require whitening of phenotype vector on each permutation (our method, only diagonalizes the covariance).

References:

Peper, J. S., Brouwer, R. M., Boomsma, D. I., Kahn, R. S. and Hulshoff Pol, H. E. (2007). Genetic influences on human brain structure: A review of brain imaging studies in twins. *Hum. Brain Mapp.*, 28: 464-473. doi:[10.1002/hbm.20398](https://doi.org/10.1002/hbm.20398)

Glahn, D. C., Thompson, P. M. and Blangero, J. (2007). Neuroimaging endophenotypes: Strategies for finding genes influencing brain structure and function. *Hum. Brain Mapp.*, 28: 488-501. doi:[10.1002/hbm.20401](https://doi.org/10.1002/hbm.20401)

Abney, M. (2015), Permutation Testing in the Presence of Polygenic Variation. *Genet. Epidemiol.*, 39: 249-258. doi:[10.1002/gepi.21893](https://doi.org/10.1002/gepi.21893)

Reviewer #1 (Remarks to the Author):

Thank you for addressing the reviewer's comments.

Reviewer #2 (Remarks to the Author):

I am happy with the responses to the reviews

We have made the requested queries. Below we summarised them:

1. Individual figures were provided.
2. Supplementary figures citation were fixed.
3. We fixed the competing interests.
4. We would like to choose OPT IN to publish the reviewers report.